# A Study on the Prediction of Cancer Using Whole-Genome Data and Deep Learning

**DOI:** 10.3390/ijms231810396

**Published:** 2022-09-08

**Authors:** Young-Ji Lee, Jun-Hyung Park, Seung-Ho Lee

**Affiliations:** 1Department of Electronic Engineering, Hanbat National University, Daejeon 34158, Korea; 23BIGS Co., Ltd., 156, Gwanggyo-ro, Yeongtong-gu, Suwon-si 16506, Korea

**Keywords:** personal genome data, genomic variation, cancers, TCGA dataset, data classification, data screening, data transformation, k-max pooling, deep learning, shortcut connection

## Abstract

The number of patients diagnosed with cancer continues to increasingly rise, and has nearly doubled in 20 years. Therefore, predicting cancer occurrence has a significant impact on reducing medical costs, and preventing cancer early can increase survival rates. In the data preprocessing step, since individual genome data are used as input data, they are classified as individual genome data. Subsequently, data embedding is performed in character units, so that it can be used in deep learning. In the deep learning network schema, using preprocessed data, a character-based deep learning network learns the correlation between individual feature data and predicts cancer occurrence. To evaluate the objective reliability of the method proposed in this study, various networks published in other studies were compared and evaluated using the TCGA dataset. As a result of comparing various networks published in other studies using the same data, excellent results were obtained in terms of accuracy, sensitivity, and specificity. Thus, the superiority of the effectiveness of deep learning networks in predicting cancer occurrence using individual whole-genome data was demonstrated. From the results of the confusion matrix, the validity of the model for predicting the cancer using an individual’s whole-genome data and the deep learning proposed in this study was proven. In addition, the AUC, which is the area under the ROC curve, which judges the efficiency of diagnosis as a performance evaluation index of the model, was found to be 90% or more, good classification results were derived. The objectives of this study were to use individual genome data for 12 cancers as input data to analyze the whole genome pattern, and to not separately use reference genome sequence data of normal individuals. In addition, several mutation types, including SNV, DEL, and INS, were applied.

## 1. Introduction

Currently, the number of patients diagnosed with cancer continues to increase, and the WHO announced that the total number of people diagnosed with cancer has nearly doubled from 10 million in 2000 to 19.3 million in 2020 [1]. In 2020, there were about 10 million deaths from cancer [2]. In addition, due to the increase in the aged population, the proportion of cancer deaths among the total deaths is continuously increasing. Predicting cancer occurrence plays a significant role in reducing healthcare costs, and preventing cancer early also increases survival rates. Therefore, the bio-healthcare market, which can check an individual’s health status, has expanded worldwide, and its recent aim is to improve the quality of medical services and reduce cost [3,4]. On the other hand, although there are many causes of cancer, it is known that the probability of occurrence of cancer is remarkably high when genetic and chromosomal mutations occur. Genes are vital for the growth and development of the human body and maintaining life. However, when a small part of a gene is deleted or abnormal, it leads to various abnormalities, diseases, and death in some cases [5,6]. With the advancement of technology, the study of which genes are related to which diseases has yielded insightful results, and research on mutated genes and genetic mutations related to specific diseases continues to evolve [7,8,9]. Most cancers are caused by gene mutations; gene mutations caused by external factors (smoking, UV rays, environmental changes, etc.) are the most common cancer development pattern [4,10]. Representative mutations were revealed by research include mutated genes, such as BRCA1 and BRCA2, that can initiate the development of breast cancer [11,12]. If a mutation occurs in these genes, women have 60–80% increased risk of breast cancer and 40% increased risk of ovarian cancer [13]. In the case of men, it is reported that the risk of prostate cancer and male breast cancer is increased, and it is also associated with pancreatic cancer and colorectal cancer. Thus, research on the relationship between genes and diseases is important, and is steadily progressing. Since mutations have unique characteristics depending on the cause of their occurrence, not only the pattern analysis of specific genes, but also the pattern analysis of the whole genome is needed. However, there are many difficulties in analyzing the patterns of thousands of genes. Recently, research into extracting results from various and complex data using deep learning techniques [14,15,16,17,18,19,20] is in progress. Um et al. [21] in 2017 measured the accuracy of hand tremors by applying wearable data sensors on the hands of people with Parkinson’s disease. Similarly, Wei Jiao et al. [22] in 2020 predicted cancer using gene frequency data and gene mutation data, and Youngji Lee et al. [23] in 2020, predicted disease through a deep learning network by performing multiple data augmentations on individual microbiome data. A study by Sun et al. [24] predicted cancer through a deep learning network for 12 types of cancer based on reference genome sequence data [25] and specific genome mutation data for normal individuals. Due to the difficulty in using large training datasets, the learning procedure was limited to specific genes related to specific diseases. However, since the genome as well as mutations are unique for each person, it is possible to contract a disease even if the specific gene is not mutated. For this reason, it is necessary to analyze all gene patterns possessed by each cancer patient, rather than analyzing the gene pattern targeting only a specific gene related to a specific cancer.

Therefore, in this study, we propose a method to predict cancer using an individual’s whole-genome data (whole-exome sequencing and whole-genome sequencing) and deep learning. The objectives of the proposed study were as follows: First is the use of individual genome data for 12 cancers as input data to analyze the whole genome pattern without analyzing the genome pattern related to a specific cancer. Secondly, since the whole genome pattern of an individual is analyzed, the reference genome sequence data of the normal person were not used separately because the reference genome sequence of the normal person is not compared. Thirdly, to apply not only gene mutation types, such as SNV, DEL, and INS of point mutations, but also general genetic mutation types. Fourth, to achieve learning based on the character convolution network [26,27,28], so it can be easily applied to various diseases and rare diseases. The final objective was to analyze not only specific genomic mutations related to specific diseases, but also overall genomic patterns.

## 2. Results

### 2.1. Experiment Environment

To evaluate the objective performance of individual genomic data using the deep learning techniques suggested in this study and predicting cancer outbreaks, we use public genomic data from the Cancer Genome Atlas (TCGA) project conducted by the U.S. National Institutes of Health (NIH) [29]. The hardware used in the experiments in this study is the Intel(R) Core(TM) i7-9700K 3.60 GHz CPU, it consists of 16 GB of RAM, Nvidia GeForce RTX 2080 Ti (V-RAM 11 GB) GPU, and development tools on the Window 10 Pro 64-bit operating system used JetBrains PyCharm Community Edition 2021.1.1. In addition, torch 1.2.0, CUDA 10.0, cuDNN 7.6.5 libraries were used.

### 2.2. Experimental Procedure

After preprocessing data by performing data classification, data screening, and data transformation, training sets were used to train the deep learning network. In the learning process, backpropagation was repeatedly performed until the loss was very small to correct the weights to improve the accuracy. Next, based on the trained deep learning model, cancer prediction was performed with the untrained test set as the input.

### 2.3. Dataset

In this study, the National Cancer Institution Genomic Data Commons (NCI GDC) dataset used in this paper consisted of TCGA pan-cancer somatic mutation data [30]. The whole genome pattern was analyzed using individual genome data from 12 cancers such as BLCA, BRCA, COAD, GBM, KIRC, LGG, LUSC, OV, PRAD, SKCM, THCA, and UCEC as inputs. In addition, SNVs (SNV, DEL, and INS) were applied, and gender information was also included. Data from a total of 5554 patients were used, of which data from 4438 patients were used as the training set. In total, 892 patients were used as the validation set to determine whether learning progressed well, and 224 patients were used as the test set to confirm the learning results. Table 1 shows the names and numbers of datasets of 12 cancer types. Figure 1 shows an example of the test set used for evaluation.

### 2.4. Reliability Evaluation of Cancer Prediction Results

In this study, it took approximately 11 s to load the training model and process the 224 sources of test data. To evaluate the objective reliability of the prediction result of cancer using individual’s whole-genome data and the deep learning technique proposed in this study, the accuracy, precision, specificity, sensitivity, F-score, etc., were comparatively evaluated. In a study by Sun et al. in 2019, based on the normal gene sequence, the authors generated data by writing different sections as 1 and the same sections as 0 in the base sequence of a specific gene, and then used deep learning to identify 12 cancers. Therefore, in this study, cancer was predicted using the genetic data for 12 cancers, same as in the study by Sun et al. [24]. The 12 predicted cancers were BLCA, BRCA, COAD, GBM, KIRC, LGG, LUSC, OV, PRAD, SKCM, THCA, and UCEC. Figure 2 illustrates how the performance of the model was evaluated. In Figure 2, TP (True Positive) is when the actual value is 1 and the model predicts it as 1, and TN (True Negative) is when the actual value is 0 and the model predicts it as 0. In addition, FP (False Positive) is a case in which the actual value is 0 but the model predicts 1, and FN (False Negative) is a case in which the actual value is 1 but the model predicts 0.

Accuracy is a representative indicator used and indicates the degree to which the value to be measured is correctly predicted, and is obtained by Equation (1).
(1)$$\mathrm{Accuracy}=\frac{\mathrm{TP}+\mathrm{TN}}{\mathrm{TN}+\mathrm{FN}+\mathrm{TP}+\mathrm{FP}}$$

Sensitivity represents a case in which a predictive result is positive among people with a disease, and is obtained by Equation (2).
(2)$$\mathrm{Sensitivity}=\frac{\mathrm{TP}}{\mathrm{FN}+\mathrm{TP}}$$

Specificity indicates a case in which the prediction result is negative among people without disease, and is obtained by Equation (3).
(3)$$\mathrm{Specificity}=\frac{\mathrm{TN}}{\mathrm{FP}+\mathrm{TN}}$$

Precision represents a case in which a person with an actual disease is represented among the cases in which the prediction result is judged to be positive, and is obtained by Equation (4).
(4)$$\mathrm{Precision}=\frac{\mathrm{TP}}{\mathrm{TP}+\mathrm{FP}}$$

F-score is an index that combines precision and sensitivity, and is obtained by Equation (5).
(5)$$F–\mathrm{score}=2\times \frac{\mathrm{Precision}\times \mathrm{Sensitivity}}{\mathrm{Precision}+\mathrm{Sensitivity}}$$

The performance evaluation of this study had an accuracy of 74.11%. As shown in Table 2, the mean and standard deviation of precision were 75.7% and 14.53% respectively. The mean and standard deviation of sensitivity were 73.84 and 11.22% respectively. The mean and standard deviation of specificity were 97.61% and 1.86% respectively. The mean and standard deviation of F-score were 74.1% and 10.27% respectively. The reasons that the sensitivity was relatively lower than the specificity are as follows: First, there was little difference in the frequency of base mutations between cancers. Second, although the types of cancers were diverse, they were poorly derived because they exhibit common genetic mutations at the molecular level.

The confusion matrix [31] is an index used to evaluate the performance of a specific classification model, and expresses the relationship between the actual value and the value predicted by the model at a glance. In Figure 3, the vertical axis represents the actual cancer number, and the horizontal axis represents the predicted cancer number. In Figure 3, BRCA can be judged to be a cancer associated with UCEC and OV, since UCEC and OV were predicted as well. Since the prediction results of other cancers are also expressed as the related cancers, the adequacy of the model for predicting cancer using an individual’s whole-genome data and deep learning proposed in this study is proven.

In addition, Receiver Operating Characteristic (ROC) curves judge the efficiency of diagnosis as a performance evaluation index of the model [32]. The ROC curve is not only widely used for diagnostic purposes in the medical field, but is also a commonly used index for algorithm performance evaluation. The ROC curve is a graph expressing the relationship between specificity and sensitivity according to a standard, and the horizontal axis shows specificity, and the vertical axis indicates sensitivity. As the ROC curve rises to the top left, and thus the Area Under the Curve (AUC) increases, it can be judged that the prediction was better. Figure 4 presents the ROC curves for the 12 cancers predicted in this study. If the AUC of the ROC curve is more than 90%, it is generally recognized that the classification result is excellent. From the analysis, it is shown that the AUC is over 90% for all 12 cancers predicted in this study (Figure 4), indicating good classification results.

We compared our results with multiple networks using data for 12 cancers (BLCA, BRCA, COAD, GBM, KIRC, LGG, LUSC, OV, PRAD, SKCM, THCA, UCEC). As shown in Table 3, the method proposed in this paper had higher accuracy, sensitivity, and specificity than other networks. As a result of learning the same data using networks such as AlexNet, ResNet18, and ResNet34, poorer results were obtained than the network proposed in this paper [33,34]. It is shown that the network proposed in this paper produces better results than other existing networks because it learns by extracting features from the entire section by applying k-max pooling. Therefore, the superiority of the deep learning network proposed in this study to predict using individual whole-genome data and cancer was determined.

Furthermore, our results are compared with Sun et al. [24] in Table 4. Sun et al. [24] used WES tumor germline variants and somatic mutation data. Our paper used only somatic mutation data. Although the data of the papers are different, we compared them because it is a method for predicting multiple cancers with one network.

## 3. Discussion

Current causes of cancer include environmentally induced mutations (somatic mutation) and genetic mutation. As the number of cancers are increasing, the seriousness of somatic mutation is rising. Analyzing the mutation pattern in genes is required. There are many difficulties in analyzing thousands of gene patterns. Thus, research on various and complex data using deep learning is currently being conducted. However, since the genome and genome mutations are different for each person, even if you do not possess a specific gene, you can contract a disease. Rather than predicting with data limited to a specific genome, by learning and predicting with the whole-genome data of an individual, a more comprehensive prediction can be performed. For this reason, it is necessary to analyze all gene patterns possessed by each cancer patient, rather than analyzing the gene pattern targeting only a specific gene related to a specific cancer.

To evaluate the objective reliability of the method proposed in this study, the accuracy, precision, specificity, and F-score, were comparatively evaluated. We preformed this study using the whole-genome data for 12 cancers. The accuracy, precision, sensitivity, specificity, and F-score were 74.11%, 75.7%, 73.84%, 97.61%, and 74.1%, respectively. These results are higher than those produce by other networks. Thus, the effectiveness of our network is proven. In this study, since the reference genome sequence data for normal individuals were not needed, only individual genome data for 12 cancers were used as training data to analyze an individual’s whole-genome mutation pattern. The training data includes several mutation types includes SNV, DEL, and INS. It also includes characteristic information about chromosome number, position, and type of genetic variation. The whole-genome data used in this paper consisted of genome sequencing data and exome sequencing data, and information on genomic mutations occurring between the start and end positions of genome mutations is specified. Therefore, it was judged that the information that affects genomic variation, such as missense, silent, frame shift, etc., is necessary for learning. As learning is performed based on the character convolution network, it can be easily applied to various diseases, including rare diseases. Therefore, it is possible to analyze not only specific genome mutations, but also the whole genome pattern. The preprocessed character-based one-hot encoding vector extracts feature information through a convolution layer and a pooling layer. Therefore, overfitting does not occur because there is a greater possibility of extracting feature information that affects the result rather than feature information on repeated characters. The proposed method uses individual genome data for 12 cancers as input data to analyze the whole genome pattern, and does not separately use reference genome sequence data for normal individuals.

On the other hand, our results were expressed as a confusion matrix to evaluate the performance between cancers related to the prediction results of a specific cancer predicted by the model. In addition, the classification results were shown as the AUC, the area under the ROC curve, which is an index for evaluating the performance of algorithms, and these results were more than 90%; therefore, the ROC curve is shaped well for the 12 predicted cancers.

As a result of comparing the method proposed in this study using the same data for various networks published in other studies, excellent results were obtained in terms of accuracy, sensitivity, and specificity. Thus, the superiority of deep learning networks in predicting cancer using individual whole-genome data is demonstrated.

In summary, the evaluation of the prediction of cancer using whole-genome data and deep learning proposed in this study is as follows. First, it is considered as a basis for predicting various types of cancer because it does not learn specific genetic data related to specific cancers, but analyzes all genome patterns of each cancer patient. Second, the reason that the results for uterine cancer were lower than those of other cancers is that BRCA1 and BRCA2 mutant genes are related to both breast and uterine cancer. Third, this study did not use reference sequence information for normal people and predicts cancer using only individual genomic data information. Given the flexibility of this approach, it can be extended to other diseases as well. As a future research plan, it is considered necessary to apply genomic data not only to cancer, but also other diseases, including rare diseases.

## 4. Materials and Methods

The proposed network predicts 12 types of cancer through a deep learning network using individual whole-genome data. Figure 5 presents the overall schema of the prediction of cancer using whole-genome data and deep learning. The data used in the proposed method are individual whole-genome data. Since genomic data has various mutation information for each chromosome, the type of word increases exponentially as the quantity of data, including new genomic mutation information, increases. On the other hand, since the number of characters is specified for a character, the number of characters does not change, even if the quantity of data increases. Therefore, after embedding data in character units, the results are derived through a deep learning network.

### 4.1. Overall Flow Chart of This Study

The overall flow chart of this study consists of a data preprocessing stage that processes data classification, data screening, data transformation, and the deep learning network, which learns using training sets among the preprocessed data, as shown in Figure 6. After data preprocessing is performed, genome data are learned using the deep learning network based on character convolution. In the execution process, cancer prediction is performed using test sets based on the learned model.

### 4.2. Data Preprocessing

The data for 12 cancers, BLCA, BRCA, COAD, GBM, KIRC, LGG, LUSC, OV, PRAD, SKCM, THCA, and UCEC, among the published genomic data of the TCGA project conducted by the National Institutes of Health [35], were included in this study. In total, 80% of the total 5554 data were in the training set, and the rest were divided to a validation set and test set. The data preprocessing stage consists of a data classification process that classifies individual data, a data screening process that reduces training time for the network results, and a data transformation process that converts into a learning-enabled form.

#### 4.2.1. Data Classification

Individual whole-genome data were used as input data. However, since TCGA data were not classified by individual, they were classified as individual whole-genome data using Sample Barcode. The Sample Barcode is a unique key for classifying each biological sample data in the TCGA database. Figure 7 shows the classification of TCGA data into individual whole-genome data using Sample Barcode.

#### 4.2.2. Data Screening

After classifying into individual whole-genome data, data screening is performed. Data screening is an important process that increases accuracy and performance of the network result by excluding unclear and unrelated features. First, TCGA genetic data holds a variety of information, but environmental and acquired factors such as tobacco, weight, alcohol, etc., were excluded from variable information in this study because there was not enough data provided due to personal information protection. Six categories of the TCGA data variables are selected, and five categories for variables excluding cancer types are used as training data. Variables used in training data are chromosome number, chromosome location, genetic mutation, mutation type, and gender. The mutation type is an important variable in identifying genetic mutations because it contains information about whether the amino acid sequence changes when the sequence changes. Figure 8 shows the variable information to be used for learning data.

#### 4.2.3. Data Transformation

After the data selection, this process performs data transformation into a form that can be learned by the deep learning network. Various individual genomic data are converted into one sentence in order to apply to a character convolution-based deep learning network. As shown in Figure 9, each column represents different mutation feature information. Each row represents mutation feature information separated by ‘ ’(spacing), and each row is separated by a ‘.’(dot).

After converting multiple genomic data of an individual into a single sentence, uppercase letters are changed to lowercase letters and one-hot encoding is performed. Since the training data is the whole-genome data for each individual, there are several mutations in the genome of each individual. Therefore, even when new genome mutation information comes in, one-hot encoding is performed that converts data into character units that are easy to generate and learn data. Using a, b, c, d, e, f, g, h, I, j, k, l, m, n, o, p, q, r, s, t, u, v, w, x, y, z, 0, 1, 2, 3, 4, 5, 6, 7, 8, 9, -, ’, ;, etc.(70 characters) for characters convolution, the sentence of “hi hello” is performed one-hot encoding with 1 for the corresponding character and 0 for the non-corresponding character as in the example of Figure 10.

Based on the selected TCGA genome data variable information, one-hot encoding is per-formed to build 4438 training data as shown in Figure 11. The average number of variants for each patient was 217. The constructed training data is given as input data to deep learning network based on character convolution.

### 4.3. Deep Learning Network Architecture

Based on the selected TCGA genome data variable information, the constructed training data goes through the character convolution neural network shown in Figure 12, and the results of predicting cancer using genomic data are the output. It has been proven by previously published studies that when the layer of the deep learning network model is too deep, the performance of the model is lowered due to problems such as gradient vanishing and gradient exploding [36,37]. Gradient vanishing is a case in which the weights are not properly updated as the gradient gradually decreases toward the input layer in the backpropagation process during deep model training. Gradient exploding refers to a case in which the gradient increases and diverges to an abnormally large value. To solve this problem, the ResNet model for learning deep layers has been published [34]. In this study, we use shortcut connection, which is the basic schema of the ResNet model. Skipping several layers through a shortcut connection that skips more than one layer can solve the gradient vanishing problem and improve the learning speed. On the other hand, by using batch normalization that normalizes the distribution of data, the dependence of the initial weight value is reduced, and overfitting, which decreases the error for training data but increases the error for actual data, is reduced.

Convolutional neural networks generally consist of a convolution layer and a pooling layer, and a fully connected layer [38] is applied at the last stage. The convolution layer extracts various features from the input data, and the pooling layer uses both max pooling and k-max pooling to prevent overfitting by reducing the size and parameters of the input data. Max pooling is down sampling by extracting the maximum value for each period of data. K-max pooling [39] performs down sampling by extracting k with the highest value from each row of learning data. Since the dataset used in this study is sequencing data, even though there are important data values in a certain section, there is a possibility of loss if it is not the maximum value. Therefore, k-max pooling is performed before the fully connected layer to extract features from the whole section. After the pooling layer, features are classified using a fully connected layer. Finally, the learning time is improved using the ReLu activation function [40], and the probability of cancer prediction results is estimated by performing softmax.

#### Backpropagation

In this study, backpropagation [41,42] is used to reduce the loss when the loss rises higher than certain level. The effect of improving the accuracy of cancer prediction results was obtained. After comparing the results obtained by the deep learning network and the results of the validation set, the backpropagation is repeated until the loss is very small to obtain the desired result for the output data. The weights are modified so that the data used for training is the genome data of a total of 4438 people, and the data used in the validation set are the genome data of 892 people. Epoch was set to 150 times, batch size was set to 8, and learning rate was set to 0.0001.

## Figures and Tables

**Figure 1 ijms-23-10396-f001:**
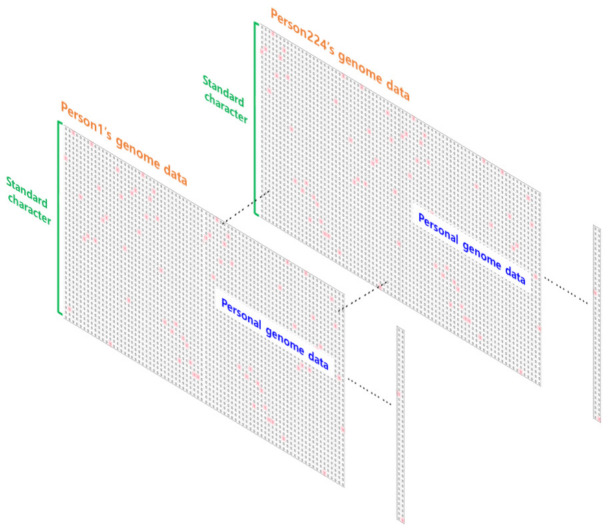
Example of a sample dataset used for evaluation.

**Figure 2 ijms-23-10396-f002:**
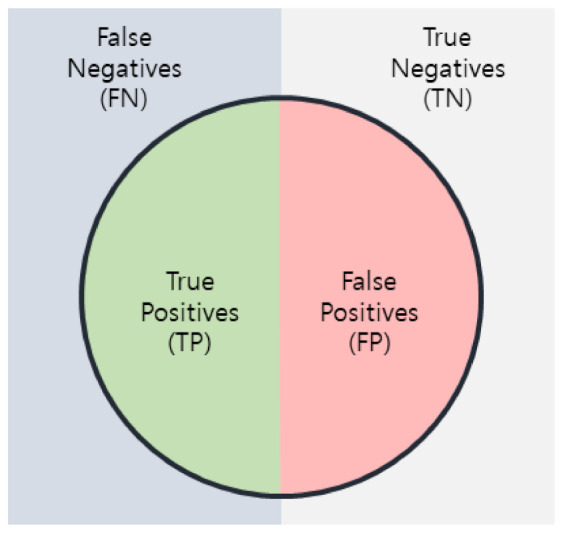
Performance evaluation index of the model.

**Figure 3 ijms-23-10396-f003:**
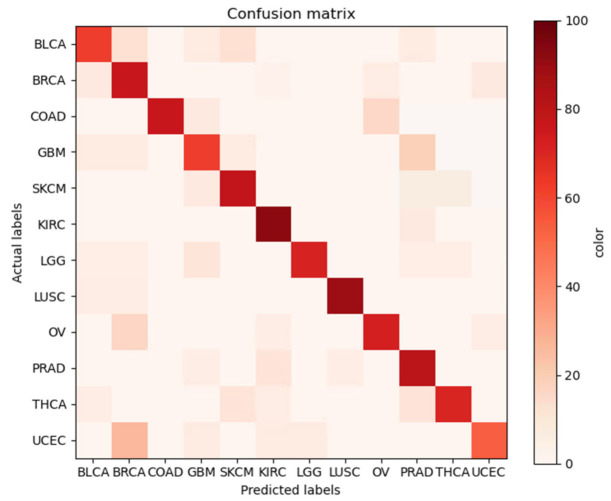
Confusion matrix of 12 cancers.

**Figure 4 ijms-23-10396-f004:**
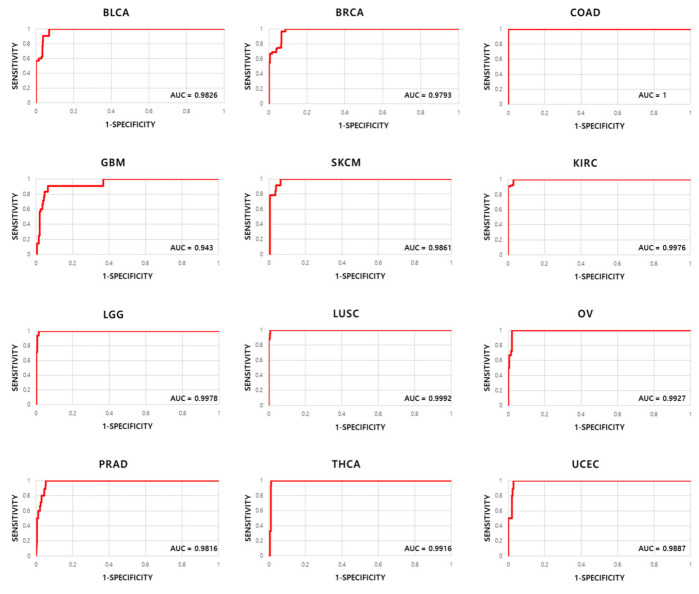
ROC curves of 12 cancers.

**Figure 5 ijms-23-10396-f005:**
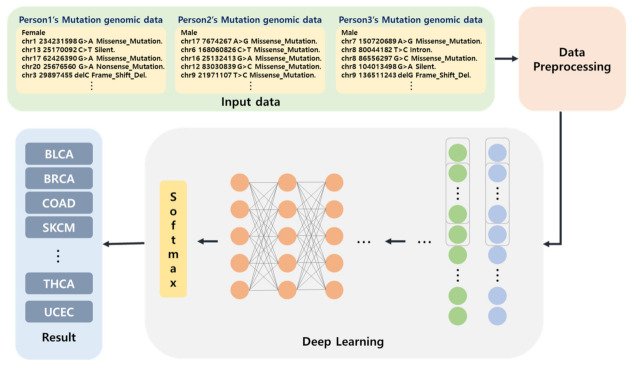
The overall schema for the prediction of cancer using whole-genome data and deep learning.

**Figure 6 ijms-23-10396-f006:**
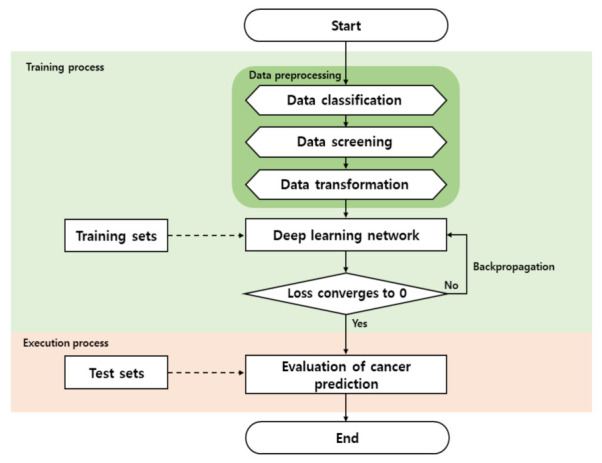
The overall flow chart of this study.

**Figure 7 ijms-23-10396-f007:**
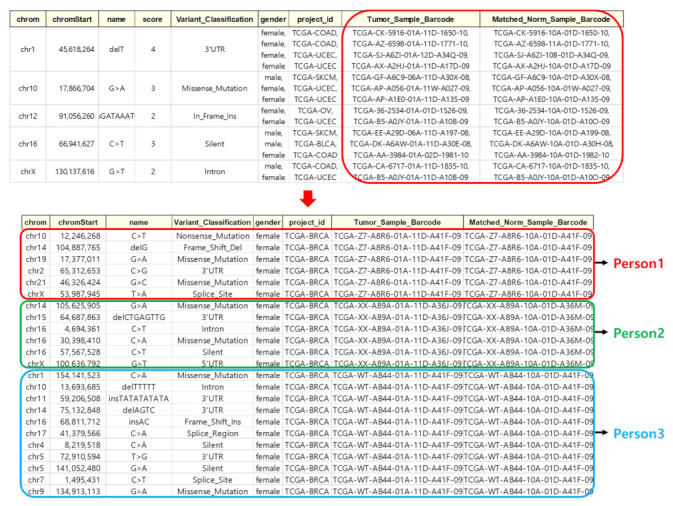
Classified by individual genomic data.

**Figure 8 ijms-23-10396-f008:**
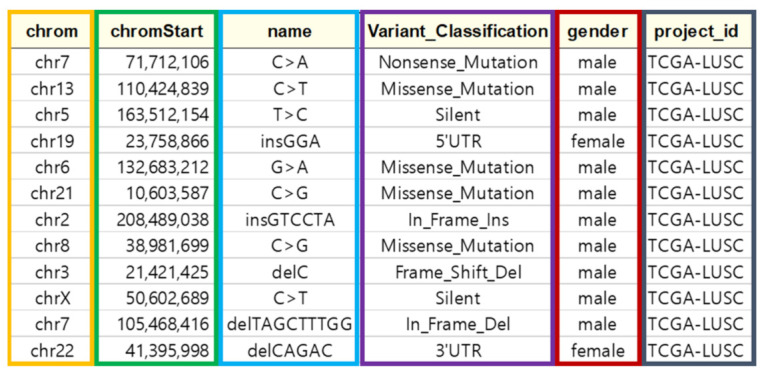
Variable information to be used as training data.

**Figure 9 ijms-23-10396-f009:**
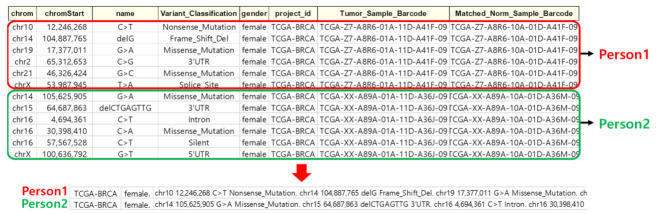
The process of converting an individual’s genome data into a single sentence.

**Figure 10 ijms-23-10396-f010:**
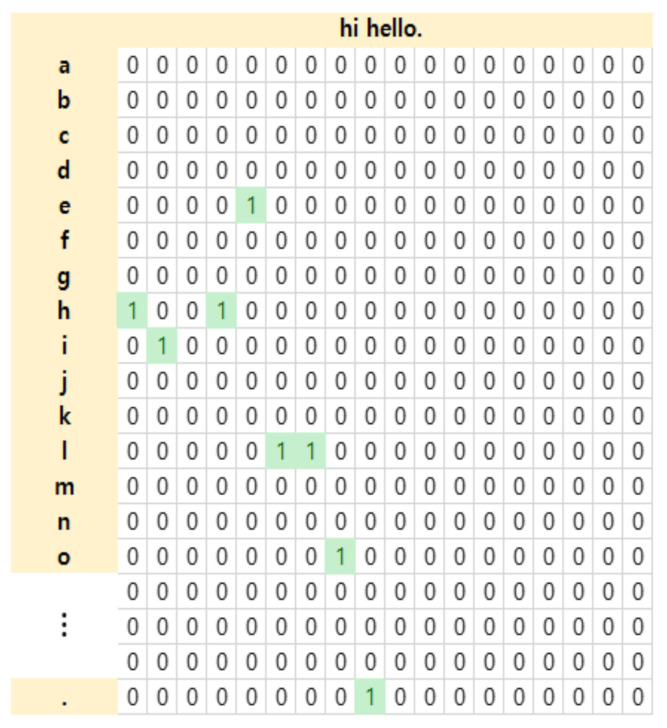
Example of one-hot encoding.

**Figure 11 ijms-23-10396-f011:**
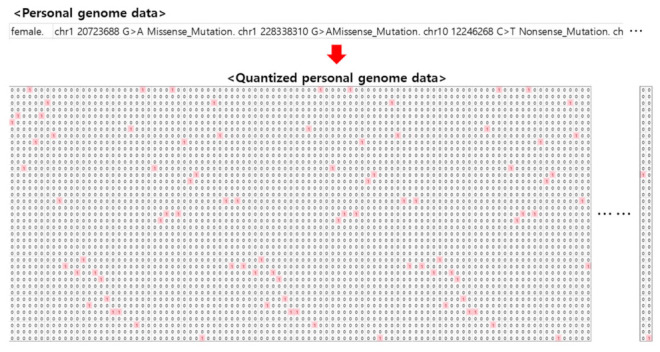
Construction of learning data after data conversion for personal genome data.

**Figure 12 ijms-23-10396-f012:**
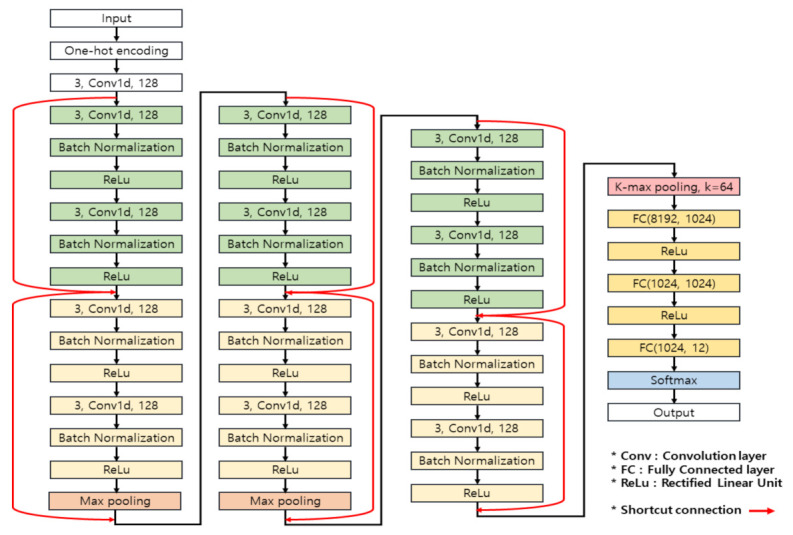
Structure of the character convolution neural network applied in this study.

**Table 1 ijms-23-10396-t001:** Name and number of dataset for 12 cancers.

Label	Cancer Name	Cancer	Training Set	Validation Set	Test Set	Total
1	Bladder urothelial carcinoma	BLCA	314	63	16	393
2	Breast invasive carcinoma	BRCA	776	157	38	971
3	Colon adenocarcinoma	COAD	258	52	13	323
4	Glioblastoma multiforme	GBM	304	61	16	381
5	Skin cutaneous melanoma	SKCM	285	58	14	357
6	Kidney renal clear cell carcinoma	KIRC	268	53	14	335
7	Brain lower-grade glioma	LGG	405	81	21	507
8	Lung squamous cell carcinoma	LUSC	379	76	19	474
9	Ovarian serous cystadenocarcinoma	OV	344	68	18	430
10	Prostate adenocarcinoma	PRAD	394	79	20	493
11	Thyroid carcinoma	THCA	393	79	20	492
12	Uterine corpus endometrial carcinoma	UCEC	318	65	15	398
Total	12	12	4438	892	224	5554

**Table 2 ijms-23-10396-t002:** Results of precision, sensitivity, specificity, and F-score for 12 cancers.

Label	Cancer Name	Cancer	Test Set	Precision	Sensitivity	Specificity	F-Score
1	Bladder urothelial carcinoma	BLCA	16	58.8	62.5	96.63	60.6
2	Breast invasive carcinoma	BRCA	38	70.7	76.3	93.55	73.4
3	Colon adenocarcinoma	COAD	13	100.0	76.9	100.0	87.0
4	Glioblastoma multiforme	GBM	16	58.8	62.5	96.63	60.6
5	Skin cutaneous melanoma	SKCM	14	68.8	78.6	97.62	73.3
6	Kidney renal clear cell carcinoma	KIRC	14	68.4	92.9	97.14	78.8
7	Brain lower-grade glioma	LGG	21	93.8	71.4	99.51	81.1
8	Lung squamous cell carcinoma	LUSC	19	94.4	89.5	99.51	91.9
9	Ovarian serous cystadenocarcinoma	OV	18	76.5	72.2	98.06	74.3
10	Prostate adenocarcinoma	PRAD	20	64	80.0	95.59	71.1
11	Thyroid carcinoma	THCA	20	87.5	70.0	99.02	77.8
12	Uterine corpus endometrial carcinoma	UCEC	15	66.7	53.3	98.09	59.3
Average				75.7	73.84	97.61	74.1

**Table 3 ijms-23-10396-t003:** Results of proposed method and other networks.

Method	Accuracy	Sensitivity	Specificity
AlexNet [33]	56.68	56.89	95.62
ResNet18 [34]	66.07	64.95	96.84
ResNet34 [34]	67.69	66.21	96.99
The Proposed Method	74.11	73.84	97.61

**Table 4 ijms-23-10396-t004:** Results of the proposed method and those of Sun et al. [24].

Method	Accuracy	Sensitivity	Specificity
Sun et al. [24] (using the WES tumor germline variants and somatic mutation data)	70.4	65.92	96.27
The Proposed Method (using only somatic mutation data)	74.11	73.84	97.61

## Data Availability

The dataset supporting the conclusions of this article is available for download at the https://tcga-data.nci.nih.gov/docs/publications/tcga/, accessed on 31 July 2022. All source code developed by this study is publicly available at: https://github.com/leeyoungji/Prediction-Cancer-using-Whole-Genome-Data, accessed on 31 July 2022.

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
