# Peer review of "A Study on the Prediction of Cancer Using Whole-Genome Data and Deep Learning"

_ijms, 2022, doi:10.3390/ijms231810396_

Round 1

Reviewer 1 Report

In this manuscript, the authors investigate the utility of deep learning in the prediction of the association between whole genome data and cancer. The authors have uploaded the code to GitHub and have also described in detail the dependencies in which the code will be run in the paper. The tables and figures show the details of the experiment well. All the reference list is sufficient to support this study.

There are some weak points in the manuscripts which need to be improved:

With regard to the results of precision, sensitivity, specificity, and F-score for 12 cancers, variance or standard deviation could be added. As we can see from the table, the results vary quite a lot from one cancer to another and the mean may not describe these differences very well.

Author Response

Point 1: With regard to the results of precision, sensitivity, specificity, and F-score for 12 cancers, variance or standard deviation could be added. As we can see from the table, the results vary quite a lot from one cancer to another and the mean may not describe these differences very well.

Response 1: The standard deviation for each evaluation index in this paper is as follows. We have added that information to 2.4 Reliability evaluation of cancer prediction results: “The performance evaluation of this study has accuracy of 74.11%. As shown in Table 2, the mean and standard deviation of precision are 75.7% and 14.53% respectively. The mean and standard deviation of sensitivity are 73.84 and 11.22% respectively. The mean and standard deviation of specificity are 97.61% and 1.86% respectively. The mean and standard deviation of F-score are 74.1% and 10.27% respectively.” [lines 142-146]

Evaluation Index

Standard Deviation

precision

14.531095

sensitivity

11.229058

specificity

1.8600788

F-score

10.276805

Reviewer 2 Report

In the paper entitled “A Study on the Prediction of Association between Whole Genome Data and Cancer using Deep Learning”, Lee and colleagues present a deep learning approach to distinguish type of cancer starting from TCGA whole genome data of 12 different cancer types.

The paper is a reproduction of the Sun paper “Identification of 12 cancer types through genome deep learning.” (Sci Rep. 2019 Nov 21;9(1):17256.) but using a different data encoding and deep learning model. The paper structure, sections and figures are very similar to Sun’s paper. There are several papers that study this topic: PMID 34689757, PMID 31911545, etc.

Deep learning modeling may have the potential to discover cross-modality correlations of multifactorial input data to delineate the interplay between tumor and the whole body. But, this paper is a methodological deep learning exercise which has not a real translational relevance. TCGA data are obtained from specimen extracted from different tissues (breast, spleen, etc). It is impossible that breast cancer is diagnosed from a bladder sample because it does not contain all the somatic mutations of cancerous cells in the breast tissue. Moreover, these methods lack of an explainability step to understand if the features that drive algorithms are in accord with biological/medical issues.

The paper is well written but the methods section must be improved with more informations about data used in the paper because there are some inaccuracies.

These are my comments:

Major comments

-  It is not clear which TCGA data have been used in this paper. Did you use germline mutations or somatic mutations? Or both? In the introduction you mentioned both BRCA variants which are germline mutation and which present in all the cells of the body but you mentioned also uv rays or smoking derived  mutations which are somatic variants. Please specify in the text. 

It is not clear which TCGA data have been used in this paper. Did you use germline mutations or somatic mutations? Or both? In the introduction you mentioned both BRCA variants which are germline mutation and which present in all the cells of the body but you mentioned also uv rays or smoking derived  mutations which are somatic variants. Please specify in the text.

How many variants did you collect on average for each patient?  Because your method of one hot encoding is affected by this number. Please specify in the text.

Character based encoding is agnostic because it is not based on biological feature of the tumors but it could be affected by the characters contained in the variables that you extracted from TCGA data. For instance the word “chr” is redundant respect to a 0 or 1 in the genomic position. Moreover, a nucleotide change in a position correspond distinctively to a certain variant classification, in this context also missense mutation, silent, or Frame shift are redundant words. This situation could be a cause of overfitting in the deep learning model? Please explain this issue in the discussion.

In my opinion these results are strongly overfitted to TCGA data. Your one hot encoding approach is based on TCGA results annotation. Unfortunately, variant calling uses different annotation such as genome version (hg19, hg38) or variant classification definition. What would happen if you will perform your algorithm trained in TCGA data with other results? It could be useful an external test cohort to validate your method

Which is the computational performance of your method? Please add it to the text.

Minor revision

- In the whole paper you wrote “Whole Genome Data” which implies the use of nucleotide variations from both introns than exons of the genome. Moreover, you indicated that you  used “ genetic data for cancers, same as the study by Sun et al” that are Whole exome sequencing data. If you used exonic data please remove “whole genome” in the text and use “exome” or the more general word “genomic”.

- At line 103 you wrote “In addition, not only point mutations but also gene mutation types such as SNP, DEL, and INS are applied”. SNPs are point mutations. Please correct in the text,

In the paper you mentioned that you did not use reference genome sequence data of normal individuals, but variant calling performed TCGA consortium use reference genome.

In the whole paper you wrote “Whole Genome Data” which implies the use of nucleotide variations from both introns than exons of the genome. Moreover, you indicated that you  used “ genetic data for cancers, same as the study by Sun et al” that are Whole exome sequencing data. If you used exonic data please remove “whole genome” in the text and use “exome” or the more general word “genomic”.

Author Response

Point 1: It is not clear which TCGA data have been used in this paper. Did you use germline mutations or somatic mutations? Or both? In the introduction you mentioned both BRCA variants which are germline mutation and which present in all the cells of the body but you mentioned also uv rays or smoking derived  mutations which are somatic variants. Please specify in the text.

Response 1: The TCGA data that we used are somatic mutations. We added that information to 2.3 Dataset: “The National Cancer Institution Genomic Data Commons (NCI GDC) data set used in this paper consists of TCGA Pan-Cancer somatic mutation data [42].” [lines 101-102]

Point 2: How many variants did you collect on average for each patient?  Because your method of one hot encoding is affected by this number. Please specify in the text.

Response 2: We are learning data from genomic mutation information that corresponds to which genomic mutation occurred between the start and the end chromosome position in a one-hot encoding method. Therefore, it is judged that genomic mutation information has a greater effect than the effect on the number of variants because data composed of mutation information for the corresponding cancer is one-hot-encoded and learned. The average number of variants in the patients we used was 217. We added that information to 4.2.3 Data transformation: “The average number of variants for each patient was 217.” [lines 336-337]

Point 3: Character based encoding is agnostic because it is not based on biological feature of the tumors but it could be affected by the characters contained in the variables that you extracted from TCGA data. For instance the word “chr” is redundant respect to a 0 or 1 in the genomic position. Moreover, a nucleotide change in a position correspond distinctively to a certain variant classification, in this context also missense mutation, silent, or Frame shift are redundant words. This situation could be a cause of overfitting in the deep learning model? Please explain this issue in the discussion.

Response 3: We used a data set consisting of genome sequencing data and exome sequencing data. The data used is composed of genome mutation information that occurred between the start and end positions of mutations. Therefore, missense, silent, frame shift etc. are not considered unnecessary information because they are information that affects genome mutation. The word "chr" may be considered an unnecessary word, but it is believed that overfitting does not occur for the following reasons. After the character-based one-hot encoding process used in this paper, the one-dimensional vector extracts feature information through the convolution layer and pooling layer shown in Fig. 12. Since the word "chr" is a one-hot encoding vector that is repeated in all data and is judged as an unnecessary feature, overfitting does not occur because the feature is not transmitted to the next layer.

We added the following to the Discussion:

1) "The whole genome data used in this paper consists of genome sequencing data and exome sequencing data, and information on genomic mutations occurring between the start and end positions of genome mutations is specified. Therefore, it is judged that information that affects genomic variation such as missense, silent, frame shift, etc. is necessary for learning." [lines 227-230]

2) "The pre-processed character-based one-hot encoding vector extracts feature information through a convolution layer and a pooling layer. Therefore, overfitting does not occur because there is a greater possibility of extracting feature information that affects the result rather than feature information on repeated characters. [lines 233-236]

Point 4: In my opinion these results are strongly overfitted to TCGA data. Your one hot encoding approach is based on TCGA results annotation. Unfortunately, variant calling uses different annotation such as genome version (hg19, hg38) or variant classification definition. What would happen if you will perform your algorithm trained in TCGA data with other results? It could be useful an external test cohort to validate your method

Response 4: The data we used is data extracted from TCGA, using global cancer patient data built and shared by the Global Cancer Big Data Platform (NCI GDC). In order to analyze and compare the algorithm trained with TCGA data from a specific hospital or cancer center, there are the following difficult problems :

1) It is very difficult to get cooperation from a specific hospital or cancer center.

2)  In order to conduct research on humans, it takes a lot of time and effort because it must be confirmed by the Institutional Review Boards (IRB), bioethics review committee.

Therefore, it is difficult to receive and analyze cancer patient data from a specific hospital or cancer center and perform comparison within a short time. However, we can tell you that in the future, we will focus a lot of effort on applying the technology developed in this paper to the actual data of domestic and foreign hospitals.

Point 5: Which is the computational performance of your method? Please add it to the text.

Response 5: The method of this paper takes about 11 seconds to load the trained model and process 224 test data. We added the following to 2.4 Reliability evaluation of cancer prediction results : “In this paper, it took approximately 11 seconds to load the training model and process the 224 test data.” [lines 117-118]

Point 6: In the whole paper you wrote “Whole Genome Data” which implies the use of nucleotide variations from both introns than exons of the genome. Moreover, you indicated that you  used “ genetic data for cancers, same as the study by Sun et al” that are Whole exome sequencing data. If you used exonic data please remove “whole genome” in the text and use “exome” or the more general word “genomic”. 

Response 6: In the TCGA website (https://docs.gdc.cancer.gov/Data/Bioinformatics_Pipelines/DNA_Seq_Variant_Calling_Pipeline/), it is explained that the data we used includes both whole exome sequencing (WES) and whole genome sequencing (WGS).

Therefore, we used the word “Whole Genome Data”. We added the information to the introduction: “whole genome data (whole exome sequencing and whole genome sequencing)” [line 72]

Point 7: At line 103 you wrote “In addition, not only point mutations but also gene mutation types such as SNP, DEL, and INS are applied”. SNPs are point mutations. Please correct in the text.

Response 7: As you pointed out, we revised the introduction as following:

Original sentence:

“In addition, not only point mutations but also gene mutation types such as SNP, DEL, and INS are applied”

Revised sentence:

 “SNPs (SNP, DEL and INS) applied”. [lines 104-105]

Point 8: In the paper you mentioned that you did not use reference genome sequence data of normal individuals, but variant calling performed TCGA consortium use reference genome.

 Response 8: For the data used in this paper, only TCGA data aligned with the human reference genome GRCh38.d1.vd1 were used. Therefore, the data used in this paper does not compare normal data with mutant data, so it does not require data of normal individuals separately.

Round 2

Reviewer 2 Report

After replies of authors, I revised the paper and, in my opinion, the design of the paper is not correct. Sun's paper used boh germline and somatic mutations “The WES tumor germline variants and somatic mutations were from twelve cancer types including BLCA, BRCA, COAD, GBM, KIRC, LGG, LUSC, OV, PRAD, SKCM, THCA and UCEC” while you used only somatic ones.

 In this context this paper remains a deep learning excercise with no translational potential. The biological question is very poor because it isn’t a method for diagnosis in fact TCGA somatic mutations derived from specimens collected in different tissues of origin (breast, bladder, spleen, etc) and it is impossible to diagnose breast cancer in spleen sample using these types of mutations, because somatic mutations are present online in cancerous cells. Moreover, the comparison between your and Sun’s results is not correct because they are performed starting from different data.

The biological terminology has several inaccuracies that could misleading.

From wikipedia: in genetics, a single-nucleotide polymorphism (SNP) is a germline substitution of a single nucleotide at a specific position in the genome. But you used somatic mutations that are not germline mutations. The general term is SNV, single nucleotide variant.

Moreover you used the term association in the title that is a genetic issue: Genetic association is when one or more genotypes within a population co-occur with a phenotypic trait more often than would be expected by chance.

You don’t use germline variants and you don’t perform any test.

You changed the last sentence in the abstract adding “other general genetic mutation types are applied”: this could imply the use also of large genomic changes (inversions, translocations, etc) or copy number variations.

Author Response

Point 1: After replies of authors, I revised the paper and, in my opinion, the design of the paper is not correct. Sun's paper used boh germline and somatic mutations “The WES tumor germline variants and somatic mutations were from twelve cancer types including BLCA, BRCA, COAD, GBM, KIRC, LGG, LUSC, OV, PRAD, SKCM, THCA and UCEC” while you used only somatic ones.

Response 1: Thanks for the reviewer's accurate point.

As pointed out by the reviewer, we used somatic mutation data, which is NCI GDC (TCGA Pan-Cancer) data. We knew that Sun's paper used tumor tissues from TCGA and both germline and somatic mutations.

The main idea of our paper is a method of predicting cancer without reference data using the individual's entire somatic mutation information. Therefore, we did not use germline data. However, it was difficult to find comparative papers that classify various cancers into one deep learning network model. Therefore, we compared the results of our paper with Sun's paper, which can be compared similarly.

We have added that information to 2.4 Reliability evaluation of cancer prediction results: “On the other hand, Sun et al.[24] used the WES tumor germline variants and somatic mutation data. However our paper used only somatic mutation data. Although the contents of the data to be compared are different, it was compared with  Sun et al.[24] because it is a method of predicting various cancers with a single learned network.”[line 190-193]

Point 2: In this context this paper remains a deep learning excercise with no translational potential. The biological question is very poor because it isn’t a method for diagnosis in fact TCGA somatic mutations derived from specimens collected in different tissues of origin (breast, bladder, spleen, etc) and it is impossible to diagnose breast cancer in spleen sample using these types of mutations, because somatic mutations are present online in cancerous cells.

Response 2: Thanks for the reviewer's accurate point

In general, it is common to analyze cancer of a specific tissue using the tissue cells.

However, because TCGA analyzed the WholeGenome sequencing data, it was able to identify various SNVs. In breast cancer, the germline BRCA1 gene is a useful marker.

However, data analysis was performed because mutations in the EGFR gene can predict metastasis to other cancers or find another marker. Therefore, it is judged that the cancer prediction process using AI, which is the result of our paper, can be continuously upgraded through actual cancer data in the future.

Meanwhile, when predicting cancer using deep learning, TCGA public data is generally used for analysis and testing. Using actual clinical data is the best method, but in general, it is too difficult to obtain cooperation with a specific hospital or cancer center to receive and analyze cancer patient data.

The TCGA Pan-Cancer data used in our paper aimed to examine the similarities and differences among the genomic and cellular alterations found across diverse tumor types. Therefore, it is judged that the cancer prediction process using AI, the result of our thesis, can be a basis for predicting the incidence rate of other cancers from one specific cancer by analyzing the similarity between cancer types.

Point 3: Moreover, the comparison between your and Sun’s results is not correct because they are performed starting from different data.

Response 3: Thanks for the reviewer's accurate point.

As pointed out by the reviewer, we used somatic mutation data, which is NCI GDC (TCGA Pan-Cancer) data. We knew that Sun's paper used tumor tissues from TCGA and both germline and somatic mutations.

The main idea of our paper is a method of predicting cancer without reference data using the individual's entire somatic mutation information. Therefore, we did not use germline data. However, it was difficult to find comparative papers that classify various cancers into one deep learning network model. Therefore, we compared the results of our paper with Sun's paper, which can be compared similarly.

We have added that information to 2.4 Reliability evaluation of cancer prediction results: “On the other hand, Sun et al.[24] used the WES tumor germline variants and somatic mutation data. However our paper used only somatic mutation data. Although the contents of the data to be compared are different, it was compared with Sun et al.[24] because it is a method of predicting various cancers with a single learned network.”[line 190-193]

Point 4: The biological terminology has several inaccuracies that could misleading.

From wikipedia: in genetics, a single-nucleotide polymorphism (SNP) is a germline substitution of a single nucleotide at a specific position in the genome. But you used somatic mutations that are not germline mutations. The general term is SNV, single nucleotide variant.

Response 4: Thanks for the reviewer's accurate point.

As the reviewer pointed out, we modified the term SNP to SNV.

  1. In addition, several mutation types includes SNV, DEL, and INS applied. [lines 27-28
  2. “Thirdly, it applies not only gene mutation types such as SNV, DEL, and INS of point mutations but also general genetic mutation types.” [lines 76-77]
  3. “The whole genome pattern is analyzed using individual genome data from 12 cancers such as BLCA, BRCA, COAD, GBM, KIRC, LGG, LUSC, OV, PRAD, SKCM, THCA, and UCEC as input. In addition, SNVs (SNV, DEL and INS) applied, and gender information is also included.” [lines 101-104]
  4. The training data includes several mutation types includes SNV, DEL, and INS. [lines 227-228]

Point 5: Moreover you used the term association in the title that is a genetic issue: Genetic association is when one or more genotypes within a population co-occur with a phenotypic trait more often than would be expected by chance.

Response 5: Thanks for the reviewer's accurate point.

As the reviewer pointed out, we revised the title.

 “A Study on the Prediction of Cancer using Whole Genome Data and Deep Learning” [line 2-3]

As pointed out by the reviewer, the sentences have been corrected as follows.

  1. Thus, the superiority of the effectiveness of deep learning networks in predicting cancer using individual whole genome data has been demonstrated.[line 19-21]
  2. In the results of the confusion matrix, the validity of the model for predicting the cancer using an individual's whole genome data and the deep learning proposed in this study was proven.[line 21-23]
  3. Therefore, in this study, we propose a method to predict cancer using an individual's whole genome data(whole exome sequencing and whole genome sequencing) and deep learning.[line 70-71]
  4. To evaluate the objective performance of individual genomic data using deep learning sug-gested in this study and predicting cancer outbreaks,[line 84-85]
  5. To evaluate the objective reliability of the prediction result of cancer using individual's whole genome data and deep learning proposed in this study,[line 117-118]
  6. the adequacy of the model for predicting cancer using an individual's whole genome data and deep learning proposed in this study has been proven.[line 156-158]
  7. Therefore, the superiority of the deep learning network proposed in this study to predict using individual whole genome data and cancer was determined.[line 198-200]
  8. Thus, the superiority of deep learning networks in predicting cancer using individual whole-genome data has been demonstrated.[line 248-249]
  9. In summary, the evaluation of the prediction of cancer using whole genome data and deep learning proposed in this study is as follows.[line 250-251]
  10. 5 is the overall schema of the prediction of cancer using whole genome data and deep learning.[line 262-263]
  11. Figure 5. prediction of cancer using whole genome data and deep learning [line 271]
  12. and the result of predicting cancer using genomic data are output.[line 343-344]

Point 6: You don’t use germline variants and you don’t perform any test.

Response 6: Thanks for the reviewer's accurate point.

The main idea of our paper is a method of predicting cancer without reference data using the individual's entire somatic mutation information. Therefore, we did not use germline data.

The reason we failed to test other data is as follows.

  1. When predicting cancer using deep learning, TCGA public data is generally used for analysis and testing. Using actual clinical data is the best method, but in general, it is too difficult to obtain cooperation with a specific hospital or cancer center to receive and analyze cancer patient data.
  2. In order to conduct research on humans, it takes a lot of time and effort because it must be confirmed by the Institutional Review Boards (IRB), bioethics review committee.

However, we can tell you that in the future, we will focus a lot of effort on applying the technology developed in this paper to the actual data of domestic and foreign hospitals.

Point 7: You changed the last sentence in the abstract adding “other general genetic mutation types are applied”: this could imply the use also of large genomic changes (inversions, translocations, etc) or copy number variations.

Response 7: Thanks for the reviewer's accurate point.

As pointed out by the reviewer, the sentences have been corrected as follows.

  1. "In addition, not only point mutation types of SNP, DEL, INS, etc., but also general genetic mutation types are applied.”
  2. “In addition, several mutation types includes SNV, DEL, and INS applied.” [line 28]
  3. “Thirdly, it applies not only gene mutation types such as SNV, DEL, and INS of point mutations but also general genetic mutation types.“
  4. “Thirdly, it applies SNVs (SNV, DEL and INS) and gender information is included.” [line 76-78]

Round 3

Reviewer 2 Report

Here, my comments:

- You replied: "In general, it is common to analyze cancer of a specific tissue using the tissue cells.

However, because TCGA analyzed the WholeGenome sequencing data, it was able to identify various SNVs. In breast cancer, the germline BRCA1 gene is a useful marker.

However, data analysis was performed because mutations in the EGFR gene can predict metastasis to other cancers or find another marker. Therefore, it is judged that the cancer prediction process using AI, which is the result of our paper, can be continuously upgraded through actual cancer data in the future."

You talked about markers like BRCA1 or EGFR, but your method has not any explainability step. It is impossible to you find new markers that dichotomized two similar tumors from different tissues.

- You put in the same table comparison data (accuracy, sensitivity, specificity) of different tests misleading the reader that are performed starting from the same data. Alexnet, ResNet18 and Resnet34 are comparisons which tested the deep learning network starting from the same data, while Sun is a sort of "external" comparison since used also germline variants.

Author Response

Point 1: You replied: "In general, it is common to analyze cancer of a specific tissue using the tissue cells.

However, because TCGA analyzed the WholeGenome sequencing data, it was able to identify various SNVs. In breast cancer, the germline BRCA1 gene is a useful marker.

However, data analysis was performed because mutations in the EGFR gene can predict metastasis to other cancers or find another marker. Therefore, it is judged that the cancer prediction process using AI, which is the result of our paper, can be continuously upgraded through actual cancer data in the future."

You talked about markers like BRCA1 or EGFR, but your method has not any explainability step. It is impossible to you find new markers that dichotomized two similar tumors from different tissues.

Response 1: Thank you very much for the kind and accurate comments of the reviewer.

Thanks for the reviewer's biologically accurate point. We didn't study how to find markers. Our research was conducted to predict cancer with AI using whole genome data. We think the results of the cancer prediction process using AI can be used as follows. By analyzing the similarity between cancer types, it can be used as a basis for predicting the incidence of other cancers in one cancer.

Point 2: You put in the same table comparison data (accuracy, sensitivity, specificity) of different tests misleading the reader that are performed starting from the same data. Alexnet, ResNet18 and Resnet34 are comparisons which tested the deep learning network starting from the same data, while Sun is a sort of "external" comparison since used also germline variants.

Response 2: Thank you very much for the kind and accurate comments of the reviewer.

As pointed out by the reviewer, the sentences have been corrected as follows.

  1. We compared our results with multiple networks using 12 cancers (BLCA, BRCA, COAD, GBM, KIRC, LGG, LUSC, OV, PRAD, SKCM, THCA, UCEC) dataset. As shown in Table 3, the method proposed in this paper has higher accuracy, sensitivity, and specificity than other networks. As a result of learning the same data using the networks such as AlexNet, ResNet18 and ResNet34, lower results were obtained than the network proposed in this paper. It is shown that the network proposed in this paper has better results than other existing networks because it learns by extracting features from the entire section by applying k-max pooling. Therefore, the superiority of the deep learning network proposed in this study to predict using individual whole genome data and cancer was determined.[line 180-188]
  2. Table 3. Results of proposed method and other networks[line 190]
  3. Also, our result was compared with Sun et al. [24] in Table 4. Sun et al.[24] used the WES tumor germline variants and somatic mutation data. Our paper used only somatic mutation data. Although the data of the papers are different, we compared them because it is a method for predicting multiple cancers with one network.[line 192-195]
  4. Table 4. Results of proposed method and one other paper[line 197]
  5. To evaluate the objective reliability of the method proposed in this study, the accuracy, precision, specificity, and F-score, were comparatively evaluated. We preformed this study using the whole genome data for 12 cancers. The accuracy, precision, sensitivity, specificity and F-score were 74.11%, 75.7%, 73.84%, 97.61% and 74.1% respectively. These results showed higher than other networks. Thus, its effectiveness has been proven.[line 211-215]
